# The Parallel Knowledge Gradient Method
# for Batch Bayesian Optimization

**Jian Wu, Peter I. Frazier**
Cornell University
Ithaca, NY, 14853
{jw926, pf98}@cornell.edu

## Abstract

In many applications of black-box optimization, one can evaluate multiple points simultaneously, e.g. when evaluating the performances of several different neural networks in a parallel computing environment. In this paper, we develop a novel batch Bayesian optimization algorithm — the parallel knowledge gradient method. By construction, this method provides the one-step Bayes optimal batch of points to sample. We provide an efficient strategy for computing this Bayes-optimal batch of points, and we demonstrate that the parallel knowledge gradient method finds global optima significantly faster than previous batch Bayesian optimization algorithms on both synthetic test functions and when tuning hyperparameters of practical machine learning algorithms, especially when function evaluations are noisy.

## 1 Introduction

In Bayesian optimization [19] (BO), we wish to optimize a derivative-free expensive-to-evaluate function $f$ with feasible domain $\mathbb{A} \subseteq \mathbb{R}^d$,

$$\min_{\boldsymbol{x} \in \mathbb{A}} f(\boldsymbol{x}),$$

with as few function evaluations as possible. In this paper, we assume that membership in the domain $\mathbb{A}$ is easy to evaluate and we can evaluate $f$ only at points in $\mathbb{A}$. We assume that evaluations of $f$ are either noise-free, or have additive independent normally distributed noise. We consider the parallel setting, in which we perform more than one simultaneous evaluation of $f$.

BO typically puts a Gaussian process prior distribution on the function $f$, updating this prior distribution with each new observation of $f$, and choosing the next point or points to evaluate by maximizing an acquisition function that quantifies the benefit of evaluating the objective as a function of where it is evaluated. In comparison with other global optimization algorithms, BO often finds "near optimal" function values with fewer evaluations [19]. As a consequence, BO is useful when function evaluation is time-consuming, such as when training and testing complex machine learning algorithms (e.g. deep neural networks) or tuning algorithms on large-scale dataset (e.g. ImageNet) [4]. Recently, BO has become popular in machine learning as it is highly effective in tuning hyperparameters of machine learning algorithms [8, 9, 19, 22].

Most previous work in BO assumes that we evaluate the objective function sequentially [13], though a few recent papers have considered parallel evaluations [3, 5, 18, 25]. While in practice, we can often evaluate several different choices in parallel, such as multiple machines can simultaneously train the machine learning algorithm with different sets of hyperparameters. In this paper, we assume that we can access $q \geq 1$ evaluations simultaneously at each iteration. Then we develop a new parallel acquisition function to guide where to evaluate next based on the decision-theoretical analysis.

**Our Contributions.** We propose a novel batch BO method which measures the information gain of evaluating $q$ points via a new acquisition function, the parallel knowledge gradient (*q-KG*). This

method is derived using a decision-theoretic analysis that chooses the set of points to evaluate next that is optimal in the average-case with respect to the posterior when there is only one batch of points remaining. Naively maximizing $q$-$KG$ would be extremely computationally intensive, especially when $q$ is large, and so, in this paper, we develop a method based on infinitesimal perturbation analysis (IPA) [25] to evaluate $q$-$KG$'s gradient efficiently, allowing its efficient optimization. In our experiments on both synthetic functions and tuning practical machine learning algorithms, $q$-$KG$ consistently finds better function values than other parallel BO algorithms, such as parallel EI [2, 19, 25], batch UCB [5] and parallel UCB with exploration [3]. $q$-$KG$ provides especially large value when function evaluations are noisy. The code in this paper is available at `https://github.com/wujian16/qKG`.

The rest of the paper is organized as follows. Section 2 reviews related work. Section 3 gives background on Gaussian processes and defines notation used later. Section 4 proposes our new acquisition function $q$-$KG$ for batch BO. Section 5 provides our computationally efficient approach to maximizing $q$-$KG$. Section 6 presents the empirical performance of $q$-$KG$ and several benchmarks on synthetic functions and real problems. Finally, Section 7 concludes the paper.

## 2   Related work

Within the past several years, the machine learning community has revisited BO [8, 9, 18, 19, 20, 22] due to its huge success in tuning hyperparameters of complex machine learning algorithms. BO algorithms consist of two components: a statistical model describing the function and an acquisition function guiding evaluations. In practice, Gaussian Process (GP) [16] is the mostly widely used statistical model due to its flexibility and tractability. Much of the literature in BO focuses on designing good acquisition functions that reach optima with as few evaluations as possible. Maximizing this acquisition function usually provides a single point to evaluate next, with common acquisition functions for sequential Bayesian optimization including probability of improvement (PI)[23], expected improvement (EI) [13], upper confidence bound (UCB) [21], entropy search (ES) [11], and knowledge gradient (KG) [17].

Recently, a few papers have extended BO to the parallel setting, aiming to choose a batch of points to evaluate next in each iteration, rather than just a single point. [10, 19] suggests parallelizing EI by iteratively constructing a batch, in each iteration adding the point with maximal single-evaluation EI averaged over the posterior distribution of previously selected points. [10] also proposes an algorithm called "constant liar", which iteratively constructs a batch of points to sample by maximizing single-evaluation while pretending that points previously added to the batch have already returned values.

There are also work extending UCB to the parallel setting. [5] proposes the GP-BUCB policy, which selects points sequentially by a UCB criterion until filling the batch. Each time one point is selected, the algorithm updates the kernel function while keeping the mean function fixed. [3] proposes an algorithm combining UCB with pure exploration, called GP-UCB-PE. In this algorithm, the first point is selected according to a UCB criterion; then the remaining points are selected to encourage the diversity of the batch. These two algorithms extending UCB do not require Monte Carlo sampling, making them fast and scalable. However, UCB criteria are usually designed to minimize cumulative regret rather than immediate regret, causing these methods to underperform in BO, where we wish to minimize simple regret.

The parallel methods above construct the batch of points in an iterative greedy fashion, optimizing some single-evaluation acquisition function while holding the other points in the batch fixed. The acquisition function we propose considers the batch of points collectively, and we choose the batch to jointly optimize this acquisition function. Other recent papers that value points collectively include [2] which optimizes the parallel EI by a closed-form formula, [15, 25], in which gradient-based methods are proposed to jointly optimize a parallel EI criterion, and [18], which proposes a parallel version of the ES algorithm and uses Monte Carlo Sampling to optimize the parallel ES acquisition function.

We compare against methods from a number of these previous papers in our numerical experiments, and demonstrate that we provide an improvement, especially in problems with noisy evaluations.

Our method is also closely related to the knowledge gradient (KG) method [7, 17] for the non-batch (sequential) setting, which chooses the Bayes-optimal point to evaluate if only one iteration is left [17], and the final solution that we choose is not restricted to be one of the points we evaluate. (Expected improvement is Bayes-optimal if the solution is restricted to be one of the points we evaluate.) We go beyond this previous work in two aspects. First, we generalize to the parallel setting.

Second, while the sequential setting allows evaluating the KG acquisition function exactly, evaluation requires Monte Carlo in the parallel setting, and so we develop more sophisticated computational techniques to optimize our acquisition function. Recently, [26] studies a nested batch knowledge gradient policy. However, they optimize over a finite discrete feasible set, where the gradient of KG does not exist. As a result, their computation of KG is much less efficient than ours. Moreover, they focus on a nesting structure from materials science not present in our setting.

## 3 Background on Gaussian processes

In this section, we state our prior on $f$, briefly discuss well known results about Gaussian processes (GP), and introduce notation used later. We put a Gaussian process prior over the function $f : \mathbb{A} \to \mathbb{R}$, which is specified by its mean function $\mu(\boldsymbol{x}) : \mathbb{A} \to \mathbb{R}$ and kernel function $K(\boldsymbol{x_1}, \boldsymbol{x_2}) : \mathbb{A} \times \mathbb{A} \to \mathbb{R}$. We assume either exact or independent normally distributed measurement errors, i.e. the evaluation $\boldsymbol{y}(\boldsymbol{x}_i)$ at point $\boldsymbol{x}_i$ satisfies

$$\boldsymbol{y}(\boldsymbol{x}_i) \mid f(\boldsymbol{x}_i) \sim \mathcal{N}(f(\boldsymbol{x}_i), \sigma^2(\boldsymbol{x}_i)),$$

where $\sigma^2 : \mathbb{A} \to \mathbb{R}^+$ is a known function describing the variance of the measurement errors. If $\sigma^2$ is not known, we can also estimate it as we do in Section 6.

Supposing we have measured $f$ at $n$ points $\boldsymbol{x}^{(1:n)} := \{\boldsymbol{x}^{(1)}, \boldsymbol{x}^{(2)}, \cdots, \boldsymbol{x}^{(n)}\}$ and obtained corresponding measurements $y^{(1:n)}$, we can then combine these observed function values with our prior to obtain a posterior distribution on $f$. This posterior distribution is still a Gaussian process with the mean function $\boldsymbol{\mu}^{(n)}$ and the kernel function $K^{(n)}$ as follows

$$\boldsymbol{\mu}^{(n)}(\boldsymbol{x}) = \mu(\boldsymbol{x})$$
$$+ K(\boldsymbol{x}, \boldsymbol{x}^{(1:n)}) \left( K(\boldsymbol{x}^{(1:n)}, \boldsymbol{x}^{(1:n)}) + \mathrm{diag}\{\sigma^2(\boldsymbol{x}^{(1)}), \cdots, \sigma^2(\boldsymbol{x}^{(n)})\} \right)^{-1} (y^{(1:n)} - \mu(\boldsymbol{x}^{(1:n)})),$$
$$K^{(n)}(\boldsymbol{x_1}, \boldsymbol{x_2}) = K(\boldsymbol{x_1}, \boldsymbol{x_2})$$
$$- K(\boldsymbol{x_1}, \boldsymbol{x}^{(1:n)}) \left( K(\boldsymbol{x}^{(1:n)}, \boldsymbol{x}^{(1:n)}) + \mathrm{diag}\{\sigma^2(\boldsymbol{x}^{(1)}), \cdots, \sigma^2(\boldsymbol{x}^{(n)})\} \right)^{-1} K(\boldsymbol{x}^{(1:n)}, \boldsymbol{x_2}).$$
$$(3.1)$$

## 4 Parallel knowledge gradient ($q$-KG)

In this section, we propose a novel parallel Bayesian optimization algorithm by generalizing the concept of the knowledge gradient from [7] to the parallel setting. The knowledge gradient policy in [7] for discrete $\mathbb{A}$ chooses the next sampling decision by maximizing the expected incremental value of a measurement, without assuming (as expected improvement does) that the point returned as the optimum must be a previously sampled point.

We now show how to compute this expected incremental value of an additional iteration in the parallel setting. Suppose that we have observed $n$ function values. If we were to stop measuring now, $\min_{x \in \mathbb{A}} \boldsymbol{\mu}^{(n)}(x)$ would be the minimum of the predictor of the GP. If instead we took one more batch of samples, $\min_{x \in \mathbb{A}} \boldsymbol{\mu}^{(n+q)}(x)$ would be the minimum of the predictor of the GP. The difference between these quantities, $\min_{x \in \mathbb{A}} \boldsymbol{\mu}^{(n)}(x) - \min_{x \in \mathbb{A}} \boldsymbol{\mu}^{(n+q)}(x)$, is the increment in expected solution quality (given the posterior after $n + q$ samples) that results from the additional batch of samples.

This increment in solution quality is random given the posterior after $n$ samples, because $\min_{x \in \mathbb{A}} \boldsymbol{\mu}^{(n+q)}(x)$ is itself a random vector due to its dependence on the outcome of the samples. We can compute the probability distribution of this difference (with more details given below), and the $q$-KG algorithm values the sampling decision $\boldsymbol{z}^{(1:q)} := \{z_1, z_2, \cdots, z_q\}$ according to its expected value, which we call the parallel knowledge gradient factor, and indicate it using the notation $q$-KG. Formally, we define the $q$-KG factor for a set of candidate points to sample $\boldsymbol{z}^{(1:q)}$ as

$$q\text{-}KG(\boldsymbol{z}^{(1:q)}, \mathbb{A}) = \min_{x \in \mathbb{A}} \boldsymbol{\mu}^{(n)}(x) - \mathbb{E}_n \left[ \min_{x \in \mathbb{A}} \boldsymbol{\mu}^{(n+q)}(x) | \boldsymbol{y}(\boldsymbol{z}^{(1:q)}) \right], \qquad (4.1)$$

where $\mathbb{E}_n \left[ \cdot \right] := \mathbb{E} \left[ \cdot | \boldsymbol{x}^{(1:n)}, y^{(1:n)} \right]$ is the expectation taken with respect to the posterior distribution after $n$ evaluations. Then we choose to evaluate the next batch of $q$ points that maximizes the parallel knowledge gradient,

$$\max_{\boldsymbol{z}^{(1:q)} \subset \mathbb{A}} q\text{-}KG(\boldsymbol{z}^{(1:q)}, \mathbb{A}). \qquad (4.2)$$

By construction, the parallel knowledge gradient policy is Bayes-optimal for minimizing the minimum of the predictor of the GP if only one decision is remaining. The *q-KG* algorithm will reduce to the parallel EI algorithm if function evaluations are noise-free and the final recommendation is restricted to the previous sampling decisions. Because under the two conditions above, the increment in expected solution quality will become

$$\min_{x \in \boldsymbol{x}^{(1:n)}} \boldsymbol{\mu}^{(n)}(x) - \min_{x \in \boldsymbol{x}^{(1:n)} \cup \boldsymbol{z}^{(1:q)}} \boldsymbol{\mu}^{(n+q)}(x) = \min y^{(1:n)} - \min \left\{ y^{(1:n)}, \min_{x \in \boldsymbol{z}^{(1:q)}} \boldsymbol{\mu}^{(n+q)}(x) \right\}$$

$$= \left( \min y^{(1:n)} - \min_{x \in \boldsymbol{z}^{(1:q)}} \boldsymbol{\mu}^{(n+q)}(x) \right)^+,$$

which is exactly the parallel EI acquisition function. However, computing *q-KG* and its gradient is very expensive. We will address the computational issues in Section 5. The full description of the *q-KG* algorithm is summarized as follows.

---

**Algorithm 1** The *q-KG* algorithm

---

**Require:** the number of initial stage samples $I$, and the number of main stage sampling iterations $N$.

1: Initial Stage: draw $I$ initial samples from a latin hypercube design in $\mathbb{A}$, $\boldsymbol{x}^{(i)}$ for $i = 1, \ldots, I$ .
2: Main Stange:
3: **for** $s = 1$ to $N$ **do**
4:   Solve (4.2), i.e. get $(z_1^*, z_2^*, \cdots, z_q^*) = \operatorname{argmax}_{\boldsymbol{z}^{(1:q)} \subset \mathbb{A}} q\text{-}KG(\boldsymbol{z}^{(1:q)}, \mathbb{A})$
5:   Sample these points $(z_1^*, z_2^*, \cdots, z_q^*)$, re-train the hyperparameters of the GP by MLE, and update the posterior distribution of $\hat{f}$.
6: **end for**
7: **return** $x^* = \operatorname{argmin}_{x \in \mathbb{A}} \mu^{(I+Nq)}(x)$.

---

# 5   Computation of *q-KG*

In this section, we provide the strategy to maximize *q-KG* by a gradient-based optimizer. In Section 5.1 and Section 5.2, we describe how to compute *q-KG* and its gradient when $\mathbb{A}$ is finite in (4.1). Section 5.3 describes an effective way to discretize $\mathbb{A}$ in (4.1). The readers should note that there are two $\mathbb{A}$s here, one is in (4.1) which is used to compute the *q-KG* factor given a sampling decision $\boldsymbol{z}^{(1:q)}$. The other is the feasible domain in (4.2) ($\boldsymbol{z}^{(1:q)} \subset \mathbb{A}$) that we optimize over. We are discretizing the first $\mathbb{A}$.

## 5.1   Estimating *q-KG* when $\mathbb{A}$ is finite in (4.1)

Following [7], we express $\boldsymbol{\mu}^{(n+q)}(x)$ as

$$\boldsymbol{\mu}^{(n+q)}(x) = \boldsymbol{\mu}^{(n)}(x) + K^{(n)}(x, \boldsymbol{z}^{(1:q)}) \left( K^{(n)}(\boldsymbol{z}^{(1:q)}, \boldsymbol{z}^{(1:q)}) \right.$$

$$\left. + \operatorname{diag}\{\sigma^2(\boldsymbol{z}^{(1)}), \cdots, \sigma^2(\boldsymbol{z}^{(q)})\} \right)^{-1} \left( y(\boldsymbol{z}^{(1:q)}) - \boldsymbol{\mu}^{(n)}(\boldsymbol{z}^{(1:q)}) \right).$$

Because $y(\boldsymbol{z}^{(1:q)}) - \boldsymbol{\mu}^{(n)}(\boldsymbol{z}^{(1:q)})$ is normally distributed with zero mean and covariance matrix $K^{(n)}(\boldsymbol{z}^{(1:q)}, \boldsymbol{z}^{(1:q)}) + \operatorname{diag}\{\sigma^2(\boldsymbol{z}^{(1)}), \cdots, \sigma^2(\boldsymbol{z}^{(q)})\}$ with respect to the posterior after $n$ observations, we can rewrite $\boldsymbol{\mu}^{(n+q)}(x)$ as

$$\boldsymbol{\mu}^{(n+q)}(x) = \boldsymbol{\mu}^{(n)}(x) + \tilde{\sigma}_n(x, \boldsymbol{z}^{(1:q)}) Z_q, \tag{5.1}$$

where $Z_q$ is a standard $q$-dimensional normal random vector, and

$$\tilde{\sigma}_n(x, \boldsymbol{z}^{(1:q)}) = K^{(n)}(x, \boldsymbol{z}^{(1:q)})(D^{(n)}(\boldsymbol{z}^{(1:q)})^T)^{-1},$$

where $D^{(n)}(\boldsymbol{z}^{(1:q)})$ is the Cholesky factor of the covariance matrix $K^{(n)}(\boldsymbol{z}^{(1:q)}, \boldsymbol{z}^{(1:q)}) + \operatorname{diag}\{\sigma^2(\boldsymbol{z}^{(1)}), \cdots, \sigma^2(\boldsymbol{z}^{(q)})\}$. Now we can compute the *q-KG* factor using Monte Carlo sampling when $\mathbb{A}$ is finite: we can sample $Z_q$, compute (5.1), then plug in (4.1), repeat many times and take average.

## 5.2 Estimating the gradient of $q$-KG when $\mathbb{A}$ is finite in (4.1)

In this section, we propose an unbiased estimator of the gradient of $q$-KG using IPA when $\mathbb{A}$ is finite. Accessing a stochastic gradient makes optimization much easier. By (5.1), we express $q$-KG as

$$q\text{-}KG(\boldsymbol{z}^{(1:q)}, \mathbb{A}) \quad = \quad \mathbb{E}_{Z_q}\left(g(\boldsymbol{z}^{(1:q)}, \mathbb{A}, Z_q)\right), \tag{5.2}$$

where $g = \min_{x \in \mathbb{A}} \boldsymbol{\mu}^{(n)}(x) - \min_{x \in \mathbb{A}} \left(\boldsymbol{\mu}^{(n)}(x) + \tilde{\sigma}_n(x, \boldsymbol{z}^{(1:q)}) Z_q\right)$. Under the condition that $\mu$ and $K$ are continuously differentiable, one can show that (please see the details in the supplementary materials)

$$\frac{\partial}{\partial z_{ij}} q\text{-}KG(\boldsymbol{z}^{(1:q)}, \mathbb{A}) \quad = \quad \mathbb{E}_{Z_q}\left(\frac{\partial}{\partial z_{ij}} g(\boldsymbol{z}^{(1:q)}, \mathbb{A}, Z_q)\right), \tag{5.3}$$

where $z_{ij}$ is the $j$th dimension of the $i$th point in $\boldsymbol{z}^{(1:q)}$. By the formula of $g$,

$$\frac{\partial}{\partial z_{ij}} g(\boldsymbol{z}^{(1:q)}, \mathbb{A}, Z_q) \quad = \quad \frac{\partial}{\partial z_{ij}} \boldsymbol{\mu}^{(n)}(x^*(\text{before})) - \frac{\partial}{\partial z_{ij}} \boldsymbol{\mu}^{(n)}(x^*(\text{after}))$$
$$- \frac{\partial}{\partial z_{ij}} \tilde{\sigma}_n(x^*(\text{after}), \boldsymbol{z}^{(1:q)}) Z_q$$

where $x^*(\text{before}) = \operatorname{argmin}_{x \in \mathbb{A}} \boldsymbol{\mu}^{(n)}(x)$, $x^*(\text{after}) = \operatorname{argmin}_{x \in \mathbb{A}} \left(\boldsymbol{\mu}^{(n)}(x) + \tilde{\sigma}_n(x, \boldsymbol{z}^{(1:q)}) Z_q\right)$, and

$$\frac{\partial}{\partial z_{ij}} \tilde{\sigma}_n(x^*(\text{after}), \boldsymbol{z}^{(1:q)}) \quad = \quad \left(\frac{\partial}{\partial z_{ij}} K^{(n)}(x^*(\text{after}), \boldsymbol{z}^{(1:q)})\right) (D^{(n)}(\boldsymbol{z}^{(1:q)})^T)^{-1}$$
$$- K^{(n)}(x^*(\text{after}), \boldsymbol{z}^{(1:q)})(D^{(n)}(\boldsymbol{z}^{(1:q)})^T)^{-1}$$
$$\left(\frac{\partial}{\partial z_{ij}} D^{(n)}(\boldsymbol{z}^{(1:q)})^T\right) (D^{(n)}(\boldsymbol{z}^{(1:q)})^T)^{-1}.$$

Now we can sample many times and take average to estimate the gradient of $q$-KG via (5.3). This technique is called infinitesimal perturbation analysis (IPA) in gradient estimation [14]. Since we can estimate the gradient of $q$-KG efficiently when $\mathbb{A}$ is finite, we will apply some standard gradient-based optimization algorithms, such as multi-start stochastic gradient ascent to maximize $q$-KG.

## 5.3 Approximating $q$-KG when $\mathbb{A}$ is infinite in (4.1) through discretization

We have specified how to maximize $q$-KG when $\mathbb{A}$ is finite in (4.1), but usually $\mathbb{A}$ is infinite. In this case, we will discretize $\mathbb{A}$ to approximate $q$-KG, and then maximize over the approximate $q$-KG. The discretization itself is an interesting research topic [17].

In this paper, the discrete set $\mathbb{A}_n$ is not chosen statically, but evolves over time: specifically, we suggest drawing $M$ samples from the global optima of the posterior distribution of the Gaussian process (please refer to [11, 18] for a description of this technique). This sample set, denoted by $\mathbb{A}_n^M$, is then extended by the locations of previously sampled points $\boldsymbol{x}^{(1:n)}$ and the set of candidate points $\boldsymbol{z}^{(1:q)}$. Then (4.1) can be restated as

$$q\text{-}KG(\boldsymbol{z}^{(1:q)}, \mathbb{A}_n) = \min_{x \in \mathbb{A}_n} \mu^{(n)}(x) - \mathbb{E}_n\left[\min_{x \in \mathbb{A}_n} \mu^{(n+q)}(x) | \boldsymbol{y}(\boldsymbol{z}^{(1:q)})\right], \tag{5.4}$$

where $\mathbb{A}_n = \mathbb{A}_n^M \cup \boldsymbol{x}^{(1:n)} \cup \boldsymbol{z}^{(1:q)}$. For the experimental evaluation we recompute $\mathbb{A}_n^M$ in every iteration after updating the posterior of the Gaussian process.

# 6 Numerical experiments

We conduct experiments in two different settings: the noise-free setting and the noisy setting. In both settings, we test the algorithms on well-known synthetic functions chosen from [1] and practical problems. Following previous literature [19], we use a constant mean prior and the ARD Matérn $5/2$ kernel. In the noisy setting, we assume that $\sigma^2(x)$ is constant across the domain $\mathbb{A}$, and we estimate it together with other hyperparameters in the GP using maximum likelihood estimation (MLE). We set $M = 1000$ to discretize the domain following the strategy in Section 5.3. In general, the $q$-KG

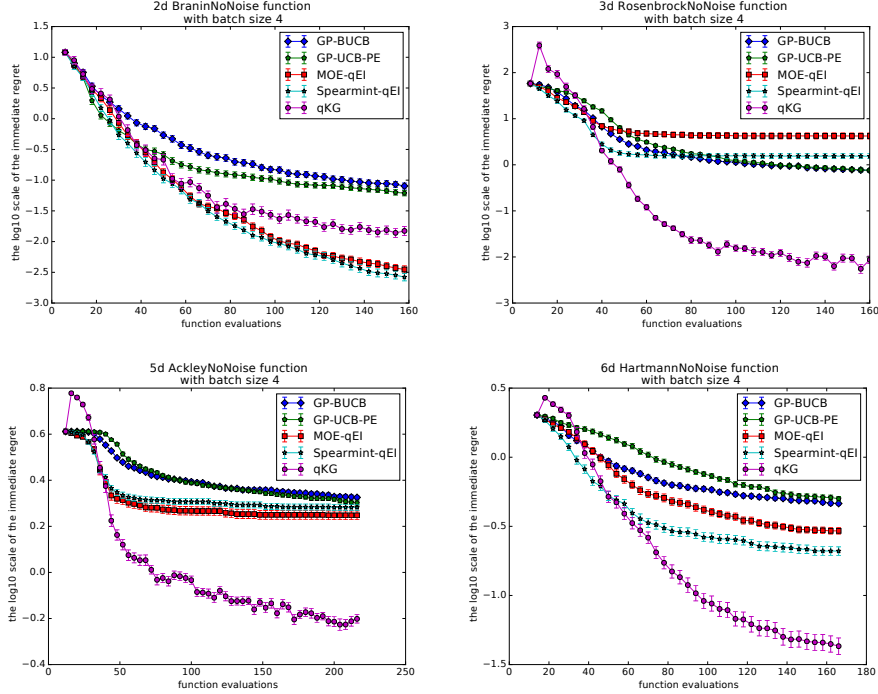

Figure 1: Performances on noise-free synthetic functions with $q = 4$. We report the mean and the standard deviation of the log10 scale of the immediate regret vs. the number of function evaluations.

algorithm performs as well or better than state-of-art benchmark algorithms on both synthetic and real problems. It performs especially well in the noisy setting.

Before describing the details of the empirical results, we highlight the implementation details of our method and the open-source implementations of the benchmark methods. Our implementation inherits the open-source implementation of parallel EI from the `Metrics Optimization Engine` [24], which is fully implemented in `C++` with a python interface. We reuse their GP regression and GP hyperparameter fitting methods and implement the *q-KG* method in `C++`. Besides comparing to parallel EI in [24], we also compare our method to a well-known heuristic parallel EI implemented in `Spearmint` [12], the parallel UCB algorithm (GP-BUCB) and parallel UCB with pure exploration (GP-UCB-PE) both implemented in `Gpoptimization` [6].

## 6.1 Noise-free problems

In this section, we focus our attention on the noise-free setting, in which we can evaluate the objective exactly. We show that parallel knowledge gradient outperforms or is competitive with state-of-art benchmarks on several well-known test functions and tuning practical machine learning algorithms.

### 6.1.1 Synthetic functions

First, we test our algorithm along with the benchmarks on 4 well-known synthetic test functions: Branin2 on the domain $[-15, 15]^2$, Rosenbrock3 on the domain $[-2, 2]^3$, Ackley5 on the domain $[-2, 2]^5$, and Hartmann6 on the domain $[0, 1]^6$. We initiate our algorithms by randomly sampling $2d + 2$ points from a Latin hypercube design, where $d$ is the dimension of the problem. Figure 3 reports the mean and the standard deviation of the base 10 logarithm of the immediate regret by running 100 random initializations with batch size $q = 4$.

The results show that *q-KG* is significantly better on Rosenbrock3, Ackley5 and Hartmann6, and is slightly worse than the best of the other benchmarks on Branin2. Especially on Rosenbrock3 and Ackley5, *q-KG* makes dramatic progress in early iterations.

### 6.1.2 Tuning logistic regression and convolutional neural networks (CNN)

In this section, we test the algorithms on two practical problems: tuning logistic regression on the MNIST dataset and tuning CNN on the CIFAR10 dataset. We set the batch size to $q = 4$.

First, we tune logistic regression on the MNIST dataset. This task is to classify handwritten digits from images, and is a 10-class classification problem. We train logistic regression on a training set with 60000 instances with a given set of hyperparameters and test it on a test set with 10000 instances. We tune 4 hyperparameters: mini batch size from 10 to 2000, training iterations from 100 to 10000, the $\ell 2$ regularization parameter from 0 to 1, and learning rate from 0 to 1. We report the mean and standard deviation of the test error for 20 independent runs. From the results, one can see that both algorithms are making progress at the initial stage while *q-KG* can maintain this progress for longer and results in a better algorithm configuration in general.

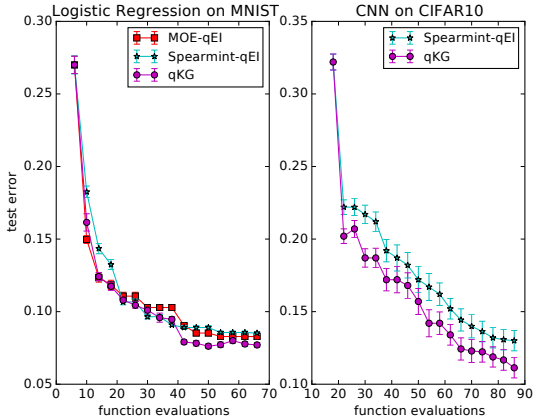

Figure 2: Performances on tuning machine learning algorithms with $q = 4$

In the second experiment, we tune a CNN on CIFAR10 dataset. This is also a 10-class classification problem. We train the CNN on the 50000 training data with certain hyperparameters and test it on the test set with 10000 instances. For the network architecture, we choose the one in `tensorflow` tutorial. It consists of 2 convolutional layers, 2 fully connected layers, and on top of them is a softmax layer for final classification. We tune totally 8 hyperparameters: the mini batch size from 10 to 1000, training epoch from 1 to 10, the $\ell 2$ regularization parameter from 0 to 1, learning rate from 0 to 1, the kernel size from 2 to 10, the number of channels in convolutional layers from 10 to 1000, the number of hidden units in fully connected layers from 100 to 1000, and the dropout rate from 0 to 1. We report the mean and standard deviation of the test error for 5 independent runs. In this example, the *q-KG* is making better (more aggressive) progress than parallel EI even in the initial stage and maintain this advantage to the end. This architecture has been carefully tuned by the human expert, and achieve a test error around 14%, and our automatic algorithm improves it to around 11%.

## 6.2 Noisy problems

In this section, we study problems with noisy function evaluations. Our results show that the performance gains over benchmark algorithms from *q-KG* evident in the noise-free setting are even larger in the noisy setting.

### 6.2.1 Noisy synthetic functions

We test on the same 4 synthetic functions from the noise-free setting, and add independent gaussian noise with standard deviation $\sigma = 0.5$ to the function evaluation. The algorithms are not given this standard deviation, and must learn it from data.

The results in Figure 4 show that *q-KG* is consistently better than or at least competitive with all competing methods. Also observe that the performance advantage of *q-KG* is larger than for noise-free problems.

### 6.2.2 Noisy logistic regression with small test sets

Testing on a large test set such as ImageNet is slow, especially when we must test many times for different hyperparameters. To speed up hyperparameter tuning, we may instead test the algorithm on a subset of the testing data to approximate the test error on the full set. We study the performance of our algorithm and benchmarks in this scenario, focusing on tuning logistic regression on MNIST. We train logistic regression on the full training set of $60,000$, but we test the algorithm by testing on $1,000$ randomly selected samples from the test set, which provides a noisy approximation of the test error on the full test set.

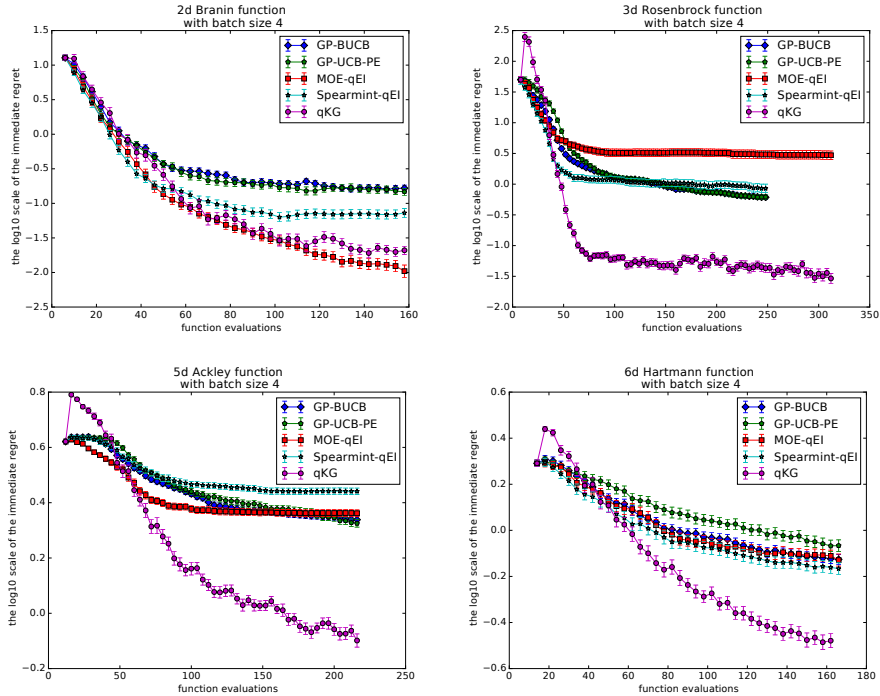

Figure 3: Performances on noisy synthetic functions with $q = 4$. We report the mean and the standard deviation of the log10 scale of the immediate regret vs. the number of function evaluations.

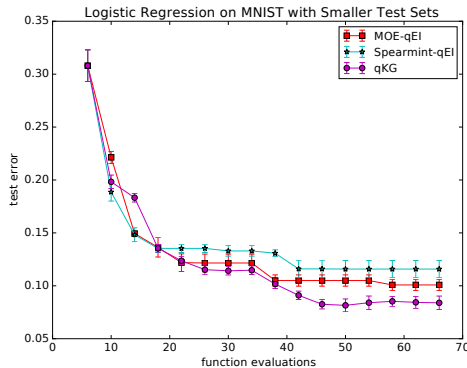

Figure 4: Tuning logistic regression on smaller test sets with $q = 4$

We report the mean and standard deviation of the test error on the full set using the hyperparameters recommended by each parallel BO algorithm for 20 independent runs. The result shows that *q-KG* is better than both versions of parallel EI, and its final test error is close to the noise-free test error (which is substantially more expensive to obtain). As we saw with synthetic test functions, *q-KG*'s performance advantage in the noisy setting is wider than in the noise-free setting.

### Acknowledgments

The authors were partially supported by NSF CAREER CMMI-1254298, NSF CMMI-1536895, NSF IIS-1247696, AFOSR FA9550-12-1-0200, AFOSR FA9550-15-1-0038, and AFOSR FA9550-16-1-0046.

## 7    Conclusions

In this paper, we introduce a novel batch Bayesian optimization method *q-KG*, derived from a decision-theoretical perspective, and develop a computational method to implement it efficiently. We show that *q-KG* outperforms or is competitive with the state-of-art benchmark algorithms on several synthetic functions and in tuning practical machine learning algorithms.

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
