[Supplementary Material]

# The Parallel Knowledge Gradient Method for Batch Bayesian Optimization Supplementary Materials

**Jian Wu, Peter I. Frazier**
Cornell University
Ithaca, NY, 14853
{jw926, pf98}@cornell.edu

## 1 Asynchronous $q$-*KG* Optimization

The (1.1) corresponds to the synchronous $q$-*KG* optimization, in which we wait for all $q$ points from our previous batch to finish before searching for a new batch of $q$ points. However, in some applications, we may wish to generate a new batch of points to evaluate next while $p(< q)$ points are still being evaluated, before we have their values. This is common in training machine learning algorithms, where different machine learning models do not necessarily finish at the same time.

$$\max_{\boldsymbol{z}^{(1:q)} \subset \mathbb{A}} q\text{-}KG(\boldsymbol{z}^{(1:q)}, \mathbb{A}). \tag{1.1}$$

We can generalize (1.1) to the asynchronous $q$-*KG* optimization. Given that $p$ points are still under evaluation, now we would like to recommend a batch of $q$ points to evaluate. As we did for the synchronous $q$-*KG* optimization above, now we estimate the $q$-*KG* of the combined $q + p$ points only with respect to the $q$ points that we need to recommend. Then we proceed the same way via gradient-based algorithms.

## 2 Speed-up analysis

Next, we compare $q$-*KG* at different levels of parallelism against the fully sequential KG algorithm. We test the algorithms with different batch sizes on two noisy synthetic functions Branin2 and Hartmann6, whose standard deviation of the noise is $\sigma = 0.5$. From the results, our parallel knowledge gradient method does provide a speed-up as $q$ goes up.

Figure 1: The performances of $q$-*KG* with different batch sizes. We report the mean and the standard deviation of the log10 scale of the immediate regret vs. the number of iterations. Iteration 0 is the initial designs. For each iteration later, we evaluate $q$ points recommended by the $q$-*KG* algorithm.

# 3 The unbiasedness of the stochastic gradient estimator

Recall that in Section 5 of the main document, we have expressed the *q-KG* factor as follows,

$$q\text{-}KG(\boldsymbol{z}^{(1:q)}, \mathbb{A}) = \mathbb{E}\left(g(\boldsymbol{z}^{(1:q)}, \mathbb{A}, Z_q)\right) \tag{3.1}$$

where the expectation is taken over $Z_q$ and

$$g(\boldsymbol{z}^{(1:q)}, \mathbb{A}, Z_q) = \min_{x \in \mathbb{A}} \boldsymbol{\mu}^{(n)}(x) - \min_{x \in \mathbb{A}} \left(\boldsymbol{\mu}^{(n)}(x) + \tilde{\sigma}_n(x, \boldsymbol{z}^{(1:q)}) Z_q\right),$$

$$\tilde{\sigma}_n(x, \boldsymbol{z}^{(1:q)}) = K^{(n)}(x, \boldsymbol{z}^{(1:q)})(D^{(n)}(\boldsymbol{z}^{(1:q)})^T)^{-1}.$$

The main purpose of this section is to prove the following proposition.

**Proposition 1.** *When $\mathbb{A}$ is finite, under the condition that $\mu$ and $K$ are continuous differentiable,*

$$\frac{\partial}{\partial z_{ij}} q\text{-}KG(\boldsymbol{z}^{(1:q)}, \mathbb{A})\bigg|_{\boldsymbol{z}^{(1:q)}=\theta^{(1:q)}} = \mathbb{E}\left(\frac{\partial}{\partial z_{ij}} g(\boldsymbol{z}^{(1:q)}, \mathbb{A}, Z_q)\right)\bigg|_{\boldsymbol{z}^{(1:q)}=\theta^{(1:q)}}, \tag{3.2}$$

*where $1 \leq i \leq q$, $1 \leq j \leq d$, $z_{ij}$ is the jth dimension of the ith point in $\boldsymbol{z}^{(1:q)}$ and $\theta^{(1:q)} \in$ the interior of $\mathbb{A}^q$.*

Without loss of generality, we assume that (1) $i$ and $j$ are fixed in advance and (2) $\mathbb{A} = [0,1]^d$, we would like to prove that (3.2) is correct. Before proceeding, we define one more notation $f_{\mathbb{A},Z_q}(z_{ij}) := g(\boldsymbol{z}^{(1:q)}, \mathbb{A}, Z_q)$ where $\boldsymbol{z}^{(1:q)}$ equals to $\theta^{(1:q)}$ component-wise except for $z_{ij}$. To prove it, we cite Theorem 1 in [1], which requires three conditions to make (3.2) valid: there exists an open neighborhood $\Theta = (0,1)$ of $\theta_{ij}$ where $\theta_{ij}$ is the jth dimension of ith point in $\theta^{(1:q)}$ such that (i) $f_{\mathbb{A},Z_q}(z_{ij})$ is continuous in $\Theta$ for any fixed $\mathbb{A}$ and $Z_q$, (ii) $f_{\mathbb{A},Z_q}(z_{ij})$ is differentiable except on a denumerable set in $\Theta$ for any given $\mathbb{A}$ and $Z_q$, (iii) the derivative of $f_{\mathbb{A},Z_q}(z_{ij})$ (when it exists) is uniformly bounded by $\Gamma(Z_q)$ for all $z_{ij} \in \Theta$, and the expectation of $\Gamma(Z_q)$ is finite.

## 3.1 The proof of condition (i)

Under the condition that the mean function $\mu$ and the kernel function $K$ are continuous differentiable, we see that for any given $x$, $\tilde{\sigma}_n(x, \boldsymbol{z}^{(1:q)})$ is continuous differentiable in $\boldsymbol{z}^{(1:q)}$ by the result that the multiplication, the inverse (when the inverse exists) and the Cholesky operators [2] preserve continuous differentiability. When $A$ is finite, we see that $g(\boldsymbol{z}^{(1:q)}, \mathbb{A}, Z_q) = \min_{x \in \mathbb{A}} \boldsymbol{\mu}^{(n)}(x) - \min_{x \in \mathbb{A}} \left(\boldsymbol{\mu}^{(n)}(x) + \tilde{\sigma}_n(x, \boldsymbol{z}^{(1:q)}) Z_q\right)$ is continuous in $\boldsymbol{z}^{(1:q)}$. Then $f_{\mathbb{A},Z_q}(z_{ij})$ is also continuous in $z_{ij}$ by the definition of the function $f_{\mathbb{A},Z_q}(z_{ij})$.

## 3.2 The proof of condition (ii)

By the expression that $f_{\mathbb{A},Z_q}(z_{ij}) = \min_{x \in \mathbb{A}} \boldsymbol{\mu}^{(n)}(x) - \min_{x \in \mathbb{A}} \left(\boldsymbol{\mu}^{(n)}(x) + \tilde{\sigma}_n(x, \boldsymbol{z}^{(1:q)}) Z_q\right)$, if both $\arg\min_{x \in \mathbb{A}} \boldsymbol{\mu}^{(n)}(x)$ and $\arg\min_{x \in \mathbb{A}} \left(\boldsymbol{\mu}^{(n)}(x) + \tilde{\sigma}_n(x, \boldsymbol{z}^{(1:q)}) Z_q\right)$ are unique, then $f_{\mathbb{A},Z_q}(z_{ij})$ is differentiable at $z_{ij}$. We define $D(\mathbb{A}) \subset \Theta$ to be the set that $f_{\mathbb{A},Z_q}(z_{ij})$ is not differentiable, then we see that

$$D(\mathbb{A}) \subset \cup_{x,x' \in \mathbb{A}} \left\{z_{ij} \in \Theta : \boldsymbol{\mu}^{(n)}(x) = \boldsymbol{\mu}^{(n)}(x'), \frac{d\boldsymbol{\mu}^{(n)}(x)}{dz_{ij}} \neq \frac{d\boldsymbol{\mu}^{(n)}(x')}{dz_{ij}}\right\} \cup$$

$$\cup_{x,x' \in \mathbb{A}} \left\{z_{ij} \in \Theta : h_x(z_{ij}) = h_{x'}(z_{ij}), \frac{dh_x(z_{ij})}{dz_{ij}} \neq \frac{dh_{x'}(z_{ij})}{dz_{ij}}\right\}$$

where $h_x(z_{ij}) := \boldsymbol{\mu}^{(n)}(x) + \tilde{\sigma}_n(x, \boldsymbol{z}^{(1:q)}) Z_q$. $\boldsymbol{\mu}^{(n)}(x)\left(\boldsymbol{\mu}^{(n)}(x')\right)$ depend on $z_{ij}$ if $x = z_i$ ($x' = z_i$) where $z_i$ is the ith point of $\boldsymbol{z}^{(1:q)}$. As $\mathbb{A}$ is finite, we only need to show that $\left\{z_{ij} \in \Theta : \boldsymbol{\mu}^{(n)}(x) = \boldsymbol{\mu}^{(n)}(x'), \frac{d\boldsymbol{\mu}^{(n)}(x)}{dz_{ij}} \neq \frac{d\boldsymbol{\mu}^{(n)}(x')}{dz_{ij}}\right\}$ and $\left\{z_{ij} \in \Theta : h_x(z_{ij}) = h_{x'}(z_{ij}), \frac{dh_x(z_{ij})}{dz_{ij}} \neq \frac{dh_{x'}(z_{ij})}{dz_{ij}}\right\}$ is denumerable.

Defining $\eta(z_{ij}) := h_{x_1}(z_{ij}) - h_{x_2}(z_{ij})$ on $\Theta$, one can see that $\eta(z_{ij})$ is continuous differentiable on $\Theta$. We would like to show that $E := \left\{z_{ij} \in \Theta : \eta(z_{ij}) = 0, \frac{d\eta(z_{ij})}{dz_{ij}} \neq 0\right\}$ is denumerable. To

prove it, we will show that $E$ contains only isolated points. Then one can use a theorem in real analysis: any set of isolated points in $\mathbb{R}$ is denumerable (see the proof of statement 4.2.25 on page 165 in [3]). To prove that $E$ only contains isolated points, we use the definition of an isolated point: $y \in E$ is an isolated point of $E$ if and only if $x \in E$ is not a limit point of $E$. We will prove by contradiction, suppose that $y \in E$ is a limit point of $E$, then it means that there exists a sequence of points $y_1, y_2, \cdots$ all belong to $E$ such that $\lim_{n \to \infty} y_n = z_{ij}$. However, by the definition of derivative and $\eta(y_n) = \eta(z_{ij}) = 0, 0 \neq \frac{d\eta(y)}{dy}\big|_{y=z_{ij}} = \lim_{n \to \infty} \frac{\eta(y_n)-\eta(z_{ij})}{y_n-z_{ij}} = \lim_{n \to \infty} 0 = 0$, a contradiction. So we conclude that $E$ only contains isolated points, so is denumerable.

Defining $\delta(z_{ij}) := \boldsymbol{\mu}^{(n)}(x_1) - \boldsymbol{\mu}^{(n)}(x_2)$ on $\Theta$, $\delta(z_{ij})$ is also continuous differentiable on $\Theta$, then one can similarly prove that $\left\{ z_{ij} \in \Theta : \delta(z_{ij}) = 0, \frac{d\delta(z_{ij})}{dz_{ij}} \neq 0 \right\}$ is denumerable.

### 3.3 The proof of condition (iii)

Recall that from Section 5 of the main document,

$$
\begin{aligned}
\frac{d}{dz_{ij}} f(z_{ij}, \mathbb{A}, Z_q) &= \frac{\partial}{\partial z_{ij}} g(\boldsymbol{z}^{(1:q)}, \mathbb{A}, Z_q) \\
&= \frac{\partial}{\partial z_{ij}} \boldsymbol{\mu}^{(n)}(x^*(\text{before})) - \frac{\partial}{\partial z_{ij}} \boldsymbol{\mu}^{(n)}(x^*(\text{after})) \\
&\quad - \frac{\partial}{\partial z_{ij}} \tilde{\sigma}_n(\boldsymbol{z}^{(1:q)}, x^*(\text{after})) Z_q,
\end{aligned}
$$

where $x^*(\text{before}) = \operatorname{argmin}_{x \in \mathbb{A}} \boldsymbol{\mu}^{(n)}(x)$, $x^*(\text{after}) = \operatorname{argmin}_{x \in \mathbb{A}} \left(\boldsymbol{\mu}^{(n)}(x) + \tilde{\sigma}_n(x, \boldsymbol{z}^{(1:q)}) Z_q\right)$, and

$$
\begin{aligned}
\frac{\partial}{\partial z_{ij}} \tilde{\sigma}_n(\boldsymbol{z}^{(1:q)}, x^*(\text{after})) &= \left( \frac{\partial}{\partial z_{ij}} K^{(n)}(x^*(\text{after}), \boldsymbol{z}^{(1:q)}) \right) (D^{(n)}(\boldsymbol{z}^{(1:q)})^T)^{-1} \\
&\quad - D^{(n)}(x^*(\text{after}), \boldsymbol{z}^{(1:q)})(D^{(n)}(\boldsymbol{z}^{(1:q)})^T)^{-1} \\
&\quad \left( \frac{\partial}{\partial z_{ij}} D^{(n)}(\boldsymbol{z}^{(1:q)})^T \right) (D^{(n)}(\boldsymbol{z}^{(1:q)})^T)^{-1}.
\end{aligned}
$$

We can calculate the $\frac{\partial}{\partial z_{ij}} \boldsymbol{\mu}^{(n)}(x)$ as follows

$$
\frac{\partial}{\partial z_{ij}} \boldsymbol{\mu}^{(n)}(x) = \begin{cases} \frac{\partial}{\partial z_{ij}} \boldsymbol{\mu}^{(n)}(z_i) & \text{if } x = z_i, \text{ i.e. the } i\text{th point of } \boldsymbol{z}^{(1:q)} \\ 0 & \text{otherwise.} \end{cases}
$$

Using the fact that $\mu$ is continuously differentiable and $\mathbb{A}$ is compact, then $\frac{\partial}{\partial z_{ij}} \boldsymbol{\mu}^{(n)}(x)$ is bounded by some $B > 0$. By the result that $\frac{\partial}{\partial z_{ij}} \tilde{\sigma}_n(\boldsymbol{z}^{(1:q)}, x^*(\text{after}))$ is continous, it is bounded by a vector $0 \leq \Lambda < \infty$ as $\mathbb{A}$ is compact. Then $\left| \frac{d}{dz_{ij}} f(z_{ij}, \mathbb{A}, Z_q) \right| \leq 2B + \sum_{i=1}^{q} \Lambda_i |z_i|$ where $Z_q = (z_1, \cdots, z_q)^T$. And $\mathbb{E}\left( \sum_{i=1}^{q} \Lambda_i |z_i| \right) = \sqrt{2/\pi} \sum_{i=1}^{q} \Lambda_i < \infty$.

## 4 The convergence of stochastic gradient ascent

In this section, we will prove that SGA converges to a stationary point. We follow the same idea of proving the Theorem 2 in [4].

First, it requires the step size $\gamma_t$ satisfying $\gamma_t \to 0$ as $t \to \infty$, $\sum_{t=0}^{\infty} \gamma_t = \infty$ and $\sum_{t=0}^{\infty} \gamma_t^2 < \infty$. Second, it requires the second moment of the gradient estimator is finite. In the above section 1.3, we have show that $|\frac{\partial}{\partial z_{ij}} g(\boldsymbol{z}^{(1:q)}, \mathbb{A}, Z_q)| \leq 2B + \sum_{i=1}^{q} \Lambda_i |z_i|$, then $\mathbb{E}(\frac{\partial}{\partial z_{ij}} g(\boldsymbol{z}^{(1:q)}, \mathbb{A}, Z_q))^2 \leq 4B^2 + \sum_{i=1}^{q} \Lambda_i^2 + 4B\sqrt{2/\pi} \sum_{i=1}^{q} \Lambda_i < \infty$.