[Reviews · NeurIPS 2016]

Reviewer 1

Summary

This paper is about batch-sequential Bayesian Optimization. A batch-sequential version of the Knowldege Gradient criterion is introduced. After some background on related work and Gaussian processes, the parallel (q-KG) criterion is defined. Its computation is detailed, following the route of the standard KG criterion, and a number of numerical experiments are presented where q-KG appears to outperform some state-of-the-art batch-sequential Bayesian Optimization algorithms.

Qualitative Assessment

This is a very good paper and I think that it possesses important qualities that would justify its publication in NIPS. The potential impact on society of parallelizing Bayesian optimization algorithms is paramount. A minor criticism is that speed-ups (from sequential to batch-sequential) are not studied, and also a few statements are slightly imprecise in the literature review; for instance the idea of integrating EI with respect to posterior distributions was already present in "Kriging is well-suited to parallelize optimization" (along with Constant Liar), while Chevalier et al. presented notably CL-mix in [2]. Also q-EI maximization using natural gradient is studied in "Differentiating the multipoint Expected Improvement for optimal batch design". Finally I have the following remarks and questions: * Is A compact, f continuous? * Why restrict A to be an LHS at the initial stage ? * In Algorithm 1: what about hyperparameter re-estimation (or Bayesian updating)? * In 5.2: is it a max or a min in the definition of g? * About 5.2 again: the smoothness of g doesn't seem to go without saying. * Similarly, the derivative of the Cholesky factor might be non obvious. * Matérn is 5/2 and not 2/5 I guess * About Figure 1 and the corresponding experiments: was it possible to fit the same GPs using different softwares?

Confidence in this Review

3-Expert (read the paper in detail, know the area, quite certain of my opinion)


Reviewer 2

Summary

Bayesian optimization has become very popular for machine learning tasks like hyperparameter optimization, but most existing approaches are inherently sequential. The paper proposes a parallel Bayesian optimization algorithm that computes Bayes-optimal batches. The authors show that the proposed method outperforms existing Bayesian optimization approaches.

Qualitative Assessment

The paper is well written and easy to follow. Parallelization of BO is an important subject for practical hyperparameter optimization and the proposed approach is interesting and more elegant than most existing approaches I am aware of. The fact a Bayes-optimal batch is determined is very promising. The authors assume independent normally distributed errors, which is common in most BO methods based on Gaussian processes. However, in hyperparameter optimization this assumption is problematic, since measurements errors represent the difference between generalization performance and empirical estimates (e.g., through cross-validation). Structural bias is common in certain regions of hyperparameter-space, especially given small sample sizes. I think it is worth mentioning that assuming independence is not always realistic. The theoretical aspect of the paper is strong, but the experiments are somewhat disappointing, for three main reasons: 1. The use of test set error as an optimization criterion is problematic, and it is well known that such score-functions are suboptimal for hyperparameter tuning (e.g., [1]). Threshold-based metrics like test set error introduce a number of problems for model selection and hyperparameter optimization, so it would be better to use metrics like area under the ROC curve or log loss. 2. The authors used the MNIST and CIFAR10 data sets as a basis of their experiments, which are indeed commonly used in benchmarks. I would have preferred to see benchmarks in HPOlib [2], which is a widely-used library specifically designed to benchmark hyperparameter optimization algorithms. 3. The amount of data sets (2) used in the benchmark is very low, and the entire benchmark is based on a single machine learning task (classification) with only two learning algorithms (logistic regression and CNN). I would have liked to see more variety in the bechmark, since it is known that some optimizers perform well for some types of learning algorithms but poorly for others. Overall, I like the paper and the theoretical contribution is definitely valuable. However, the current benchmark is too limited to be truly convincing. [1] Provost, Foster J., Tom Fawcett, and Ron Kohavi. "The case against accuracy estimation for comparing induction algorithms." ICML. Vol. 98. 1998. [2] Eggensperger, Katharina, et al. "Towards an empirical foundation for assessing bayesian optimization of hyperparameters." NIPS workshop on Bayesian Optimization in Theory and Practice. 2013.

Confidence in this Review

2-Confident (read it all; understood it all reasonably well)


Reviewer 3

Summary

The paper proposes a new version of a stochastic gradient algorithm combined with Bayesian modeling, for optimizing a multivariate objective function over a finite domain. The novel part of the algorithm is that in every iteration it identifies a number of points on which to compute the objective and this computation can be parallelized. The problem is to identify the sampling points in some optimal sense. For this purpose the authors adopt the framework of Knowledge Gradient, which estimates the expected improvement in the objective value if one more sampling point is used and based on that make a decision on whether and which sampling point to use. This approach is generalized here so that the decision is made on multiple sampling points in batch. Since this problem is computationally intractable, the authors develop an algorithm based on Monte Carlo and Infinitesimal Perturbation Analysis to estimate the relevant gradients so as to maximize the information gain over the set of candidate sampling sets. The algorithm is applied on several test problems and is shown to either outperform or seriously compete with other approaches in the literature, mainly those based on Confidence Bounds.

Qualitative Assessment

The paper is well written and potentially useful for practical applications. I am not convinced that the level of novelty is very high, since the authors expand a previously studied approach (KG) to a batch setting. This requires a more sophisticated method for optimizing the criterion over a set of multiple points. The main novelty of the paper in my opinion lies in the estimation of the q-KG gradient based on an IPA approach, which is also rather standard. I have one rather specific comments/suggestion: In equation (3.1), since x^(1:n) is a finite sequence of vectors, it is not clear how mu(x^(1:n)) and K(x(1:n),x(1:n)) are defined, since mu and K are functions of (one or two) vectors and not sequences.

Confidence in this Review

2-Confident (read it all; understood it all reasonably well)


Reviewer 4

Summary

This paper considers the parallel knowledge gradient method for batch Bayesian Optimization. The authors consider the case when evaluating the function value is time consuming. They propose a new acquisition function called q-KG that can output several new points for parallel evaluation. They also provide an efficient approach to evaluate the q-KG function. Empirical results are shown to verify their theoretical results.

Qualitative Assessment

This paper considers the parallel knowledge gradient method for batch Bayesian Optimization. The problem they considered is important. Being able to split the work in a distributed way is useful to the performance of data driven applications. Instead of evaluating one point at a time, the algorithm consider q knowledge gradient that propose q point to evaluate at next iteration, which fasten the efficiency of the algorithm a lot. It is an interesting idea to be explored. Yet it is unclear whether Gaussian process is a good assumption to make, especially in neural network architectures where the dimension of the input is very large. It might be helpful if the author could provide some theoretical results of their algorithm on the error bound of their methods and the quantitative analysis of the complexity with regard to dimension. But this is a solid work overall in the field of Bayesian Optimization. The method is solid and the experiment is sound. The paper is well written overall. Some typos and suggestions are l.36 "the set of points to evaluate next that is" l.95 What is the definition of A? l.104 What is the input of mu^n? l.128 It would be helpful to briefly discuss the parallel EI algorithm l.140 It might help to use different notations for A.

Confidence in this Review

2-Confident (read it all; understood it all reasonably well)


Reviewer 5

Summary

This paper derives a batch-version of Knowledge Gradient (KG). To find an optimal batch of design points, a gradient estimation method based on Monte-Carlo simulation is proposed. The experiments show that the proposed method outperforms other batch Bayesian optimization (BO) methods.

Qualitative Assessment

1. As I know, the batch KG is not newly proposed method by this paper. Please clarify the contribution of this paper with respect to previous batch KG papers such as [1] and [2]. In my opinion, the contribution is estimating the gradient for finding the optimal batch. If so, the efficiency/effectiveness of the proposed method should be compared with the previous batch KG papers by experiments or possibly mathematical analysis. 2. Section 5.2 should be elaborated more, since I think the section is the key point of this paper. 3. Basically, EI assumes noiseless observations. Hence, the comparison in the noisy setting is unfair. Does a variant of EI regarding noise is used? If the plain EI is used, the results in Section 6.2 should be discussed with the reason that EI does not consider the noise. 4. In line 162, the abbreviation “IPA” is redefined. [1] Yingfei Wang, Kristofer G. Reyes, Keith A. Brown, Chad A. Mirkin, and Warren B. Powell. 2015. Nested-Batch-Mode Learning and Stochastic Optimization with An Application to Sequential MultiStage Testing in Materials Science. SIAM J. Sci. Comput. 37, 3 (January 2015), B361–B381. DOI:http://dx.doi.org/10.1137/140971117 [2] http://castlelab.princeton.edu/theses/Peng%20-%20Senior%20Thesis%20-%20May032010.pdf

Confidence in this Review

3-Expert (read the paper in detail, know the area, quite certain of my opinion)


Reviewer 6

Summary

This work provides a method to perform parallel Batch Bayesian Optimization by computing a Bayes-optimal batch of configurations to evaluate next. The proposed method measures the utility of evaluating a batch of configurations using the knowledge gradient (q-KG). To efficiently maximize the acquisition function they present a method based on infinitesimal perturbation analysis to estimate the gradient of q-KG. In 3 experiments they show, that q-KG performs competitive with existing parallel Bayesian optimization methods on problems without observation noise and it outperforms these methods on problems with observation noise.

Qualitative Assessment

I do like the idea of performing parallel hyperparameter optimization to allow the efficient use of multi-core environments, but some aspects remain unclear to me and I am not convinced by the experimental results. Please find feedback for each of the points above and some questions at the end: # Technical quality The paper used common artificial benchmark function to show superiority of the proposed method. Additional to these cheap-to-evaluate problems the paper also presents results for optimizing a CNN and logistic regression for Cifar10/MNIST. In the experiment section I would like to see an experiment showing wall-clock time or an analysis on how much one can gain from using this parallel method compared to simple random search or sequential optimization. Also I do not see, why q should be set to 4 for all experiments. # Novelty/Originality The paper proposes to use the knowledge gradient (a generalization of expected improvement) as an acquisition function and infinitesimal perturbation analysis (IPA) to maximize q-KG. Both methods themselves are not new and the paper refers to "Parallel Bayesian global optimization of expensive functions", which already shows how to use IPA to select a batch of configurations using expected improvement. # Impact/Usefulness Parallel hyperparameter tunings method are highly relevant for (real-world) applications from many disciplines, such as machine learning, computer vision, biology and robotics. # Clarity/Presentation The technical details and the background section are described well and the experiment section contains everything necessary to comprehend the performed comparisons. Questions to the authors: 1.) As Wang et al. (2015) seems to be very relevant to this work and code seems to be available online, have you compared to this method? (Or is this "parallel EI in MOE", then you should call it q-EI to distinguish it from Spearmint's parallel method). To avoid confusion it would be beneficial to clearly state the differences in the paper. 2.) Nowadays multi-core systems allow for much more than 4 parallel runs. For example * "Scalable Bayesian Optimization Using Deep Neural Networks", ICML'15 used up to 800 parallel runs of DNGO * "Parallel Algorithm Configuration", LION'12 used 25 SMAC runs in parallel * "Practical Bayesian Optimization of Machine Learning Algorithms", NIPS'12 used up to 10 parallel runs of Spearmint Do you have any insights to what extent your method scales and how much one can gain from more parallel runs? 3.) Right now your method is supposed to run synchronously. Would it be possible to extend it to run asynchronously, such as for example Spearmint does? 4.) In the end, wall-clock time matters. Have you evaluated how your method compares to random search or simple sequential Bayesian optimization with respect to wall-clock time? How large is the overhead for computing a new batch? 5.) What does "iterations" as the x-label in your plots mean. Is it the number of function evaluations or number of batches? And does the x-axis include the initial design? If yes, why do not all methods start from the same value? Minor comments: * Is there a z^(1:n) missing in eq (4.1)? * "this architecture has been carefully tuned by the human expert, and achieve a test error" -> achieves * Parenthetical references are not nouns, e.g. "[16] suggests parallelizing" should be "Snoek et al. [16] suggest parallelizing". * I don't think "Wang et al. (2015)" is the correct reference for IPA, as they do not introduce the method, but use and apply it.

Confidence in this Review

2-Confident (read it all; understood it all reasonably well)